# Pickering Emulsion of Curcumin Stabilized by Cellulose Nanocrystals/Chitosan Oligosaccharide: Effect in Promoting Wound Healing

**DOI:** 10.3390/pharmaceutics16111411

**Published:** 2024-11-02

**Authors:** Long Xie, Xiaolin Dai, Yuke Li, Yi Cao, Mingyi Shi, Xiaofang Li

**Affiliations:** 1Science and Education Section, The First People’s Hospital of Shuangliu District, Chengdu (West China Airport Hospital Sichuan University), Chengdu 610299, China; xielong@stu.cdutcm.edu.cn; 2State Key Laboratory of Characteristic Chinese Medicine Resources in Southwest China, Chengdu University of Traditional Chinese Medicine, Chengdu 611137, China; daixiaolin@stu.cdutcm.edu.cn (X.D.); 2017ks310@stu.cdutcm.edu.cn (Y.L.); 2020ks310@stu.cdutcm.edu.cn (Y.C.); 3Department of pharmacy, Chengdu Seventh People’s Hospital (Affliated Cancer Hospital of Chengdu Medical College), Chengdu 610203, China; 4School of Intelligent Medicine, Chengdu University of Traditional Chinese Medicine, Chengdu 611137, China; shimingyi@cdutcm.edu.cn

**Keywords:** cellulose nanocrystals, chitosan oligosaccharide, Pickering emulsions, curcumin, wound healing

## Abstract

**Background:** The stabilization of droplets in Pickering emulsions using solid particles has garnered significant attention through various methods. Cellulose and chitin derivatives in nature offer a sustainable source of Pickering emulsion stabilizers. **Methods:** In this study, medium-chain triglycerides were used as the oil phase for the preparation of emulsion. This study explores the potential of cellulose nanocrystals (CNC) and shell oligosaccharides (COS) as effective stabilizers for achieving stable Pickering emulsions. Optical microscopy, CLSM, and Cyro-SEM were employed to analyze CNC/COS–Cur, revealing the formation of bright and uniform yellow spherical emulsions. **Results:** CLSM and SEM results confirmed that CNC/COS formed a continuous and compact shell at the oil–water interface layer, enabling a stable 2~3 microns Pickering emulsion with CNS/COS–Cur as an oil-in-water emulsion stabilizer. Based on FTIR, XRD, and SEM analyses of CNC/COS, along with zeta potential measurements of the emulsion, we found that CNC and COS complexed via electrostatic adsorption, forming irregular rods measuring approximately 200–300 nm in length. An evaluation of the DPPH radical-scavenging ability demonstrated that the CNC/ COS–Cur Pickering emulsion performed well in vitro. In vivo experiments involving full-thickness skin excision surgery in rats revealed that CNC/COS–Cur facilitated wound repair processes. Measurements of the MDA and SOD content in healing tissues indicated that the CNC/COS–Cur Pickering emulsion increased SOD levels and reduced MDA content, effectively countering oxidative stress-induced damage. An assessment based on wound-healing rates and histopathological examination showed that CNC/COS-Cur promoted granulation tissue formation, fibroblast proliferation, angiogenesis, and an accelerated re-epithelialization process within the wound tissue, leading to enhanced collagen deposition and facilitating rapid wound-healing capabilities. An antibacterial efficacy assessment conducted in vitro demonstrated antibacterial activity.

## 1. Introduction

Stabilizing droplets in Pickering emulsions requires a different method when using solid particles rather than surfactants [1]. These emulsions have numerous advantages over traditional emulsions stabilized with surfactants (e.g., low toxicity, adjustable permeability, better biocompatibility, and no added surfactants). Solid particles employed to stabilize Pickering emulsions are classified into inorganic and organic solid particles. Inorganic particles are primarily used in cosmetics, chemicals, and other applications, while organic particles are biocompatible and biodegradable [2]. Rapid advances in materials technology have expanded the range of available particles over the past few years, expanding the range of applications for Pickering emulsions. The ability of Pickering emulsions to deliver drugs has also attracted considerable attention [3]. At present, the research on Pickering emulsions in wound treatment has been initiated [4,5,6].

Cellulose is a naturally produced renewable material, extracted mainly from lignocellulosic biomass and can extract micro-particles (e.g., microcrystalline fibers and CNCs). Owing to their renewable and sustainable character, these tiny particles can be alternative biomaterials in various applications. A study validates that cellulose-based polymers can be arranged into a functional matrix that mimics the mechanical properties of healthy synovial fluid for the treatment of osteoarthritis [7]. In general, cellulose nanocrystals (CNCs) are prepared from natural cellulose through controlled sulfuric acid hydrolysis. CNCs have attracted much interest for their unique properties (e.g., replacing hyaluronic acid as a viscoelastic agent in osteoarthritis business therapy, high strength, tailored surface chemistry, biocompatibility, and sustainability). Research shows that CNCs can be used in various applications (e.g., catalysts and drug carriers, composite nano-fillers, and Pickering emulsifiers). One of the most common methods for producing negatively charged CNCs is acid hydrolysis with sulfuric acid. CNCs have a reactive surface that covers numerous hydroxyl groups. Furthermore, the surface of a CNCs obtained through sulfuric acid hydrolysis is negatively charged. Because SO_4_^2-^ counterions are attached to samples as sulfate half esters, the CNC sample gains an intrinsically negative charge [8]. However, the adsorption of nanoparticles is inhibited at the oil–water interface under the effect of the powerful repulsion between the nanoparticles. Sulfated cellulose nanocrystals (CNCs) with high surface charge densities cannot stabilize O/W emulsions, limiting their use as interfacial stabilizers [9,10].

Chitosan oligosaccharide (COS) is a chitosan oligomer prepared by deacetylating chitin. Chitin is the second-most abundant polymer in nature, found in shrimp, crab, and insect exoskeletons. Through acid hydrolysis and enzymatic degradation, chitin breaks down into COS [11]. COS exhibits antioxidant [12], anti-inflammatory [13], immunostimulatory [14], anti-obesity [14], wound-healing [15], and antibacterial [16] properties. COS is completely water soluble and less viscous due to its more limited chain lengths and free amino gatherings in D-glucosamine units [17]. Furthermore, it is bio-compatible [18], non-toxic [19], mucoadhesive [20], and non-allergenic [21]. Under the positive electrical effects of COS [22] and biological activities, it can also be employed for drug delivery [23,24].

Some studies have shown that adding an oppositely charged electrolyte to the aqueous phase supplemented with composite particles reduces the barrier effect correlated with repelling image charges. This prevents these particles from electrostatic interactions so that the composite particles can spontaneously enter the interface and stabilize the Pickering emulsion [25,26]. Cur is derived from the rhizome of the Curcuma longa plant [27]. As a functional food with several pharmacological activities, Cur is a revolutionary therapy for faster wound healing, with anti-inflammatory and free-radical-scavenging properties [28,29]. However, owing to poor aqueous solubility, the pharmaceutical uses of Cur are limited [30]. Based on this observation, we selected a complex formed via electrostatic attraction in negatively charged CNC and positively charged COS as a stabilizing agent to prepare Pickering emulsions to deliver curcumin (Cur). CNS/COS–Cur was confirmed to be an oil-in-water emulsion through characterization measurements of CNC/COS. Then, we showed that CNC/COS–Cur Pickering emulsions have better in vitro free-radical scavenging based on the results of a DPPH free-radical-scavenging ability study. The role of CNC/COS–Cur in promoting wound repair was investigated in total skin excision surgery experiments on rats. The effect of the formulation on wound healing was evaluated in terms of antioxidant properties (in vivo and in vitro), wound healing rates, and pathological sections. CNC/COS–Cur significantly enhanced rapid wound healing by promoting granulation tissue formation, fibroblast proliferation, and angiogenesis, as well as by accelerating wound tissue re-epithelialization and collagen deposition. The antibacterial properties of the emulsions were further explored in vitro.

## 2. Materials and Methods

### 2.1. Materials

#### 2.1.1. Chemicals

Curcumin (>95% purity grade) was purchased from Xi’an XiaoCao Botanical Development Co., Ltd. (Xi’an, China). CNC powder was purchased from Zhengzhou Feynman Biotechnology Co., LTD. COS powder with a molecular weight of 5 × 10 kDa was purchased from Shanghai Low Bai Biological Technology Co., LTD (Shanghai, China). Medium-chain triglyceride (MCT oil) was purchased from New Foods Co., Ltd. (Chengdu, China). All other chemical reagents used in this study were of analytical grade.

#### 2.1.2. Animals

SD male rats (180–200 g) were purchased from Chengdu Ensville Biotechnology Co., Ltd. (Chengdu, China).

### 2.2. Methods

#### 2.2.1. Preparation of the Aqueous Phase Supplemented with the CNC/COS Complex

The homogeneous dispersion of 1.2 wt% CNC solution was prepared by dissolving the CNC powder in deionized water (20 mL) and sonication for 2 min. (Type KQ5200DE CNC Ultrasonic Cleaner, Chengdu, China). COS was added to the aqueous phase in different proportions (CNC:COS—1:0.6, 1:0.8, 1:1, 1:2, 1:1.4) and stirred at 600 rpm for 120 min. (MS-280-H Magnetic Stirrer, Chengdu, China). CNC/COS complex was freeze-dried to a powder and prepared.

#### 2.2.2. Preparation of Cur Loaded Pickering Emulsion

Weighed 30 mg Cur were added in 10 mL MCT oil. Cur dissolved in the oil phase by heating and stirring at 110 °C and 800 rpm (MS-280-H Magnetic Stirrer). And the oil phase was mixed with the aqueous phase supplemented with different ratios of CNC/COS compounds so that the ratio of the oil phase to the aqueous phase was 1:2. Then, the mixture was homogenized for 8 min at 16,000 rpm using a C25 high-shear dispersion homogenizer (Shanghai Hengchuan Machinery Co., Ltd., Shanghai, China). Pickering emulsions were produced from the coarse emulsion in a Biosafer 1000 Ultrasonic Disintegrator (Biosafer (China) Co., Ltd., Wuhan, China) (ultrasound 3 s, pause 5 s). Sonication time, power, and frequency were fixed at 5 min, 300 W, and 20 kHz.

#### 2.2.3. Characterization of Cur Loaded Pickering Emulsion

##### Droplet Size and Zeta Potential Determination

The Pickering emulsion prepared in different proportions of CNC and COS, and emulsions were diluted with deionized water before analysis. Droplet sizes and zeta potential of newly prepared emulsions were determined by dynamic light scattering using a Litesizer™ 500 (Anton paar, Graz, Austria). All measurements were repeated in triplicate, from three independent samples, at 25 °C.

##### Optical Microscopy of Cur-Loaded Pickering Emulsion

Prepared Pickering emulsion droplets (CNC/COS 1:1.2) were diluted with deionized water. One drop was placed on a microscope slide. The emulsion droplets were visualized using an optical microscope (Zeiss, Oberkochen, Germany), and an image analysis software, ZEN3.0, used to analyze the emulsion droplets (Carl Zeiss Co. Ltd., Oberkochen, Germany). Micrographs of the samples were taken at 400× magnifications.

##### Confocal Laser-Scanning Microscope (CLSM)

Pickering emulsions (CNC/COS 1:1.2) were prepared by staining the oil and aqueous phases with FITC (Fluorescein isothiocyanate isomer I) and Rhodamine B, respectively (away from light). A confocal laser microscope (Olympus, Tokyo, Japan) was used to observe the microstructure of Pickering emulsions. The sample was fixed on the stage, and the plane was initially adjusted with a 60× objective lens. Fluorescence images were acquired at the excitation wavelength of 490 nm and 540 nm.

##### Cryo-Scanning Electron Microscopy (Cryo-SEM)

Pickering emulsions (CNC/COS 1:1.2) were glued to conductive carbon glue and then plunged into liquid nitrogen slush for 30 s before being transferred to the sample preparation chamber for sublimation and gold plating under vacuum using a PP3000T cryogenic frozen preparation transfer system (Quorum, UK). The samples were sublimated at −90 °C for 10 min, followed by sputter gold plating at a current of 10 mA for 60 s. Subsequently, the samples were sent to a SU3500 SEM sample chamber (Hitachi, Tokyo, Japan) for observation at a cold stage temperature of −140 °C under an accelerating voltage of 10 kv.

#### 2.2.4. Characterization of CNC/COS Complex

##### X-Ray Diffraction (XRD)

CNC/COS (1:1.2) was analyzed through XRD. The XRD patterns were recorded using an Empyrean X-ray diffractometer (Malvern PANalytica, Almelo, The Netherlands) at 40 kV and 40 mA; 2θ was set from 5° to 80°, and the scanning rate reached 2°/min.

##### Fourier Transform Infrared (FTIR) Spectroscopy

The interactions between CNC and COS were evaluated using a thermo Nicolet iS5 Fourier transform infrared spectroscopy spectrometer (Thermo Fisher, Waltham, MA, USA). The CNC/COS (1:1.2) complex powders were mixed with KBr at 1:100 and then grounded in a mortar by hand with a pestle. The grounded powders were pressed into flakes. The IR absorbency scans were analyzed in the range of 400–4000 cm^−1^ and at a resolution of 4 cm^−1^, in which air served as the background.

##### Scanning Electron Microscopy (SEM)

Surface morphology analysis of the CNC and CNC/COS (1:1.2) complexes was conducted through Tescan Mira4 scanning electron microscopy (Tescan, Brno, Czech). Appropriate amounts of the CNC and CNC/COS complexes were weighed, glued onto conductive adhesive and platinum sprayed under vacuum for 45 s using a GMC-1500 magnetron ion sputterer. The surface morphology was identified at 70 kDa/150 kDa and then photographed.

#### 2.2.5. Cur Contents in the Pickering Emulsion

##### Equipment and Chromatographic Conditions

Chromatographic analyses were conducted on a HPLC system (Agilent Technologies, Santa Clara, CA, USA). Chromatographic conditions are presented as follows. The analysis was at 1 mL/min flow rate with UV detector at 425 nm for Cur. The samples (5 μL) were injected onto a reversed-phase column (ComatexC18—250 mm × 4.6 mm, 5 μm) and then eluted with a mobile phase (acetonitrile-0.1% phosphoric acid aqueous; 55:45, *v*/*v*). The column temperature reached 30 °C. The column was equilibrated with the mobile phase for 30 min before injection.

##### Validation of the HPLC Method

The accurately measured volumes of working-standard Cur solution (200.8 μg/mL) were transferred into a series of volumetric flasks and then diluted to volume with methanol to determine a concentration range of 0.502–200.8 μg/mL. An aliquot of CNC/COS–Cur was transferred into a 50 mL volumetric flask and then completed to the final volume with methanol to determine a concentration of 203 μg/mL. The linearity, the repeatability, the precision, percentage recoveries, LOD, and LOQ were analyzed.

##### Measurement of Cur Concentration

The Pickering emulsion samples were broken using methanol to recover the encapsulated Cur. In total, 1 mL of CNC/COS–Cur was added into 9 mL of methanol, and then the emulsion was broken with an ultrasonic probe (20 kz, 300 w). After the emulsion was broken, the solution was centrifuged to separate the CNC/COS precipitate from the clarified Cur solution. After the solution was centrifuged, the supernatant was taken, and the peak area at 425 nm was examined with HPLC [31].

#### 2.2.6. Short-Term Stability of Cur-Loaded Pickering Emulsion

Existing research has suggested that Pickering emulsions have high stability [5]. In this experiment, the short-term stability of the samples was examined. The methods of Section 2.2.1 and Section 2.2.5 were used to measure the particle size, zeta-potential of the droplets, and the concentration of Cur. The PDI was examined through dynamic light scattering using the Litesizer™ 500 in a Section 2.2.3. The variations in particle size, PDI, zeta-potential, and the Cur concentration [32] of the samples were examined over a period of one week in the dark at 25 °C and 4 °C. The anti-coagulant effect of the emulsions was identified macroscopically by visual inspection. The emulsion droplets from day 1 and day 7 were also identified under a light microscope [5].

#### 2.2.7. Antioxidant Activity of Cur-Loaded Pickering Emulsion In Vitro

The antioxidant activity of Pickering emulsions of Cur was determined by evaluating its DPPH radical-scavenging activity using the previously described method with some modifications [33,34]. The samples of MCT (Oil), CNC–Oil, CNC/COS–Oil, and CNC/COS–Cur were prepared. In total, 0.5 mL CNC/COS–Cur was mixed with 4.5 mL of DPPH solution (0.2 mM). The control solution was prepared by mixing ethanol with DPPH solution (0.2 mM). For the other samples, the same method in terms of emulsions was employed. All samples were kept in the dark for 45 min at room temperature to allow the reaction between antioxidant (Cur) and free-radicals to complete. After incubation, the absorbance of all of the samples was examined at 518 nm using a UV-6100 spectrophotometer (MAPADA, Shanghai, China). The free-radical-scavenging activity of the samples was determined by Equation (1).
Radical-scavenging activity % = (ABS_0_ − ABS_t_)/ABS_0_ × 100%(1)
where ABS_0_ is the absorbance of DPPH control solution, and ABS_t_ is the absorbance of each sample.

#### 2.2.8. Release Profile of Cur from Pickering Emulsion In Vitro

The release of Cur from Pickering’s emulsion was investigated using the dialysis method. First, 2 mL of Cur loaded Pickering emulsion and 2 mL of MCT solution of Cur were transferred into the presoaked dialysis bags with a molecular cutoff value of 14 kDa. Subsequently, the bags were suspended in 100 mL of pH 7.4 PBS (supplemented with Tween 80 (0.5% *v*/*v*)) in a shaking water bath at 37 °C at a shaking speed of 120 rpm. At regular intervals, all media were removed and then replaced with new media. Three independent experiments were performed [35]. The release percentage of Cur was determined using Equation (2).
Release% = (Amount release from Cur/Total amount encapsulated in Cur) × 100%(2)

#### 2.2.9. Wound Healing In Vivo

The experiment was performed in male SD rats weighing 180–220 g. The rats were kept at a controlled temperature (23 ± 2 °C) and exposed to 12 h of alternating light and dark. They were fed pelleted food and given unlimited access to clean cages in the experiment. All animal experiments gained approval from the Experimental Animal Protection Association of Chengdu University of Traditional Chinese Medicine, in accordance with the Guidelines for the Care and Use of Laboratory Animals published by the National Institutes of Health, US.

The effect of facilitating wound recovery of Pickering’s emulsion was studied through full-thickness skin excision [6,35,36]. The rats were randomly assigned to the three groups (n = 6 per group), i.e., Group I (Pickering-emulsion-loaded Cur), Group II (Carrier group: CNC/COS–Oil), Group III (Normal saline). Full-thickness skin excision round wounds of 8 mm in diameter were created on the dorsal back of rat after anesthesia. In total, 40 μL of medication is administered daily to the wound.

##### Morphology of the Wounds

Wound contraction was an important indicator of the healing process. Accordingly, wound healing was evaluated by measuring the reduction of the wound area. Wound tissue was photographed on days 3, 6, 9, and 12. Using image J software 1.53e, the wound area was evaluated, and the healing rate of each group was calculated using Equation (3):Wound closure % = (A_0_ − A_t_)/A_0_×100%(3)
where A_0_ is the original wound area on day 0, and A_t_ is the wound area on day t post-wounding.

##### Biochemical Analysis

Wound tissues were collected from different groups on days 3, 6, and 12 after the injury. In PBS, a 20% (*w*/*v*) homogenate was prepared using a homogenizer. The crude homogenate was centrifuged at 1000 rpm for 10 min at 4 °C to pellet-down nuclei and other cell debris. The oxidative stress marker malondialdehyde (MDA) and antioxidant biomarkers superoxide dismutase (SOD) reflect the level of oxidative stress in tissues. MDA and SOD biochemical analyses were performed on the resulting supernatant. MDA and SOD Kits were purchased from Nanjing Jiancheng Co, Ltd. (Nanjing, China).

##### Histopathology

Wound tissues were collected on days 6 and 12 after the injury from different groups. The wound tissues collected were stained for HE and Masson and examined histologically.

#### 2.2.10. Antibacterial In Vitro

The following bacterial strains were used in the antimicrobial assays: *S. aureus* (ATCC 6538, Gram-positive), Escherichia coli (ATCC 23815). The antimicrobial study involved the disk diffusion method. CNC/COS–Oil and CNC/COS–Cur were as samples. Normal saline was used as the negative control. First, agar plates were inoculated with a standardized test micro-organism. Then, filter paper disks (6 mm) supplemented with the samples were placed on the agar surface. Petri dishes were incubated under suitable conditions. After incubation, the antimicrobial agent diffuses into the agar and inhibits germination and growth of the test microorganism, and the diameters of the inhibition growth zones are examined.

#### 2.2.11. Statistical Analysis

The results were expressed as mean values ± SD (standard deviation). Statistical data were analyzed by one-way analysis of variance (ANOVA), using the SPSS software version 24 (IBM). Difference at *p* < 0.05 was considered to be significant.

## 3. Results and Discussion

### 3.1. Characterization of CNC–COS Complex

#### 3.1.1. X-ray Diffraction (XRD)

The XRD results are presented in Figure 1A. XRD is capable of suggesting the crystallinity of the respective sample [37]. The CNC exhibits three distinct characteristic diffraction peaks at 2θ values (14.900°, 16.265°, 22.725°), and the crystallinity index of CNC was calculated to be 91%, which is a highly crystalline type I cellulose compound. The COS shows one characteristic diffraction peak at 2θ values (20°). The 2θ values (20.012°) of the CNC/COS complex show distinct diffraction peaks. The diffraction peaks in the plots of the complexes are consistent with the peaks of the CNC and have not changed significantly, except that the intensity of the diffraction peaks of the complexes was relatively lower and the peak width increased. Thus, a conclusion is drawn that in the preparation of the CNC/COS complexes, the crystal structure of the CNC remained unchanged, whereas only the crystallinity of the CNC complexes was reduced compared with the CNC.

#### 3.1.2. Fourier Transform Infrared (FTIR) Spectroscopy

FTIR spectroscopy has been adopted to determine the interaction among polymers. As depicted in Figure 1B, the CNC achieved -OH-stretching vibrations at 3332.47 cm^−1^, 1029.42 cm^−1^, 1053.50 cm^−1^, 1108.10 cm^−1^, and 1160.23 cm^−1^ and S=O-stretching vibrations at 1315 cm^−1^. COS displayed overlapping -NH- and -OH-stretching vibrations around 3210.04. 1565.31 cm^−1^, 1583.32 cm^−1^, and 1311.11 cm^−1^ displayed amino-stretching vibrations. At 1030.72 cm^−1^ and 1065.96 cm^−1^, it had OH-stretching vibrations. The CNC/COS showed a strong and broad connected hydroxyl-stretching vibration peak at 3347.21 cm^−1^.

The shift of the hydroxyl-stretching vibration peak in the mixture might arise from the interference of the phenolic hydroxyl groups in the CNC and COS. In the CNC/COS, the amide-stretching vibrational peaks at 1565.31 cm^−1^ and 1583.32 cm^−1^ of COS were significantly reduced, while a conjoined hydroxyl-stretching vibrational peak at 3347.21 cm^−1^ appeared, presumably suggesting the electrostatic bonding between the amino group in COS and the surface of CNC.

#### 3.1.3. Scanning Electron Microscopy (SEM)

SEM allows for a specific observation of the surface form of an object. The SEM of the CNC and CNC/COS complex are shown in Figure 1C,D. As depicted in the figures, the CNC appeared as smooth rod-like structures with a length of 200 nm and a width of 20 nm, whereas the CNC/COS complexes appeared as long strips with an uneven surface. The length of the CNC/COS complex was also around 200 nm. The particles size affected the droplet size of the emulsion. The droplet size declined with the dimensions of the particles. The particle size for Pickering stabilization should at least be one order of magnitude smaller than the desired droplet size for the preparation of a stable emulsion [38]. As revealed by the above result, CNC/COS can be adopted to stabilize at least 2 μm droplets. Smaller particles exhibit fast adsorption kinetics, and they can be quickly and accurately adsorbed into the oil–water interface, thus controlling the droplet size [39]. As indicated by the SEM results, COS aggregates on the surface of the CNC rod-like structure to form the CNC/COS complex.

### 3.2. Characterization of Cur-Loaded Pickering Emulsion

#### 3.2.1. Droplet Size and Zeta Potential Determination

From Figure 2, the size and potential of the droplet under different proportions of CNC and COS can be identified. When the CNC: COS is 1:0.6 and 1:0.8, the droplets size is identified to be large, which is correlated with the lower zeta potential. The lower the zeta potential, the weaker the shielding effect, which is not beneficial to the stability of the emulsion. When the ratio of CNC to COS is 1:1.2, the particle size is the smallest (2333 ± 176.85 nm), and the zeta potential is 33.90 ± 3.66 mv. The results were consistent with the SEM speculation, that CNC/COS complex at about 200 nm could stabilize the droplets at about 2 µm. And it can be identified that as the proportion of COS increases, the zeta potential of droplets also increases. Because the CNC obtained by sulfuric acid hydrolysis had a negative charge [8], the zeta potential of CNC-stabilized Pickering emulsion showed a negative charge [40]. However, the zeta potential of the droplets stabilized by CNC/COS complex is examined as positive charge, suggesting that it is indeed CNC/COS-complex-stabilized droplets.

#### 3.2.2. Optical Microscopy of Pickering Emulsion of Cur

The shape, size, and uniformity of droplets can be identified directly by optical microscope. As illustrated in Figure 3A, the emulsion has a bright, even, yellow color. Under microscopic observation at 400×, it is clear that the Pickering emulsions appear as homogeneous rounded droplets.

#### 3.2.3. Confocal Laser-Scanning Microscope (CLSM)

The CLSM results are shown in Figure 3C,D. The oil phase is stained with green, and the aqueous phase is stained with red. It is identified that the green oil phase is wrapped in the red aqueous phase. CNC/COS covers the oil–water interface and forms aggregates around the oil droplets, thereby forming the emulsion gels and providing space resistance to stabilize the oil droplets. This result indicates that the obtained Cur-loaded Pickering emulsion was an O/W emulsion.

#### 3.2.4. Cryo-Scanning Electron Microscopy (Cryo-SEM)

Cryo-SEM allows microscopic imaging and analysis of emulsions. The result of Cryo-SEM is shown in Figure 3B. The shell formed by continuous insoluble particles at the oil–water interface takes on critical significance to the stability of Pickering’s emulsion [41,42]. Cryo-swept electron microscopy reveals a rough and uneven surface of the emulsion droplets. A dense layer is clearly visible on the surface of the Pickering emulsion, besides some raised flakes on the surface. Accordingly, the CNC/COS forms a dense shell at the water–oil interface to stabilize the emulsion. The CNC/COS complex aggregates on the surface of the emulsion droplets to stabilize the emulsion to form regular droplets. CNC/COS provides a novel barrier for the new Pickering emulsions and prevents flocculation, coalescence, and Ostwald ripening, such that the storage stability of the emulsions can be increased.

### 3.3. Cur Contents in the Pickering Emulsion

The result of validation of the HPLC method is shown in Table 1. Cur has a good linear range within 0.502–200.8 μg/mL. The precision, repeatability, and accuracy of this method are all satisfactory. LOD and LOQ are 0.40 μg/mL and 1.02 μg/mL for Cur, respectively. In summary, the method could be used for the analysis of Cur. The concentration of Cur in the Cur-loaded Pickering emulsion was examined as 856.9 ± 29.3 μg/mL, n = 3.

### 3.4. Short Term Stability of Cur Loaded Pickering Emulsion

As depicted in Figure 4A,B, no significant change was identified in the particle size, PDI, Zeta potential, and Cur content of the emulsion over 7 days at 25 °C and 4 °C in the dark. As indicated by B–D in Figure 4, no significant changes were identified in the morphology of the emulsion droplets as identified. Moreover, no aggregation of oil droplets was identified on a macroscopic scale, and no precipitation of the oil phase was reported. The above results confirmed that the CNC/COS–Cur exhibits high stability in the short term.

### 3.5. Antioxidant Activity In Vitro of Cur Loaded Pickering Emulsion

DPPH radical-scavenging has been generally adopted to evaluate free-radical scavenging capacity [43,44]. The results of the free-radical scavenging rate are presented as follows. Both MCT (1.06% ± 0.83) and CNC (1.67% ± 1.06) have no antioxidant effect, CNC/COS–Oil (31.42% ± 0.99), CNC/COS–Cur (84.83% ± 1.07), n = 3, *p* < 0.05. CNC/COS–Oil has weak antioxidant properties due to the activity of COS. The free-radical scavenging rate of CNC/COS–Cur with the addition of Cur was as high as 84%, which shows that CNC/COS–Cur Pickering emulsion has good antioxidant properties in vitro. The powerful free-radical scavenging ability of Pickering emulsion is mainly correlated with the antioxidant capacity of Cur. CNC/COS–Oil due to the activity of COS also has some antioxidant activity.

### 3.6. In Vitro Release Profile of Cur from Pickering Emulsion

Cur is poorly soluble in water, whereas it is lipid-soluble. Emulsion refers to an ideal solubilizing strategy for lipid-soluble drugs [45,46]. Release profile for Cur and CNC/COS–Cur are shown in Figure 5A. In 14 days, Free Cur releases 10%, and CNC/COS–Cur releases approximately 55%. The release of Cur is mainly concentrated in the first 9 days. The release of Cur in CNC/COS–Cur appeared to reach a plateau at the tenth day. Cur is an extremely insoluble drug with a slow diffusion rate into water. Moreover, the CNC/COS complex serves as a shell layer surrounding the droplets, such that the release of Cur can be slowed down. The thicker the shell layer outside the droplets, the slower the release of the Cur will be. Accordingly, Cur exhibits significant controlled-release under the CNC/COS shell encapsulation and releases in significantly higher amounts than the MCT solution of Cur. The sustained availability of Cur at the wound site may be contributed to the increase in healing activity.

### 3.7. In Vivo Wound Healing

#### 3.7.1. Morphology of the Wounds

Each group’s macroscopic observations on days 3, 6, 9, and 12 after wound injury are shown in Figure 5C. Wounds are completely closed at day 12 in all three groups after full-thickness skin excision. Wound-healing rates for each group at days 3, 6, 9, and 12 are shown in Figure 5B. The wounds started to crust on the third day, and healing was significantly accelerated in Group I. The wounds in Group I healed significantly faster. The wound-healing rates in Group I reached 33% ± 0.149 (Day 3), 82% ± 0.08 (Day 6), 97% ± 0.02 (Day 9), notably higher than both Groups II 30.3% ± 0.05 (Day 3), 65.6% ± 0.08 (Day 6), 94% ± 0.02 (Day 9) and Groups III 14.0% ± 0.7 (Day 3), 65.3% ± 0.07 (Day 6), 92.5% ± 0.01 (Day 9). The preparations loaded with Cur clearly facilitated wound closure at the early stages compared with Groups II and III. On day 12, the wounds were completely closed in all groups. Moreover, the wound in Group I exhibited a higher degree of recovery. Apparently, Cur played an active role in the wound-healing process. The wound-healing potential of Cur is attributed to its biochemical effects (e.g., anti-inflammatory, anti-infectious, and antioxidant activities) [47,48,49]. Moreover, Cur can facilitate cutaneous wound healing through involvement in tissue remodeling, granulation, tissue formation, and collagen deposition [50,51]. Existing research has suggested that the application of Cur on wounds also enhances epithelial regeneration and increases fibroblast proliferation [52]. The results of wound-closure rates make a good match to the results of HE and Masson staining. Thus, Group I can significantly contribute to wound healing compared with Groups II and III. Group II had a much higher healing rate than Group III on day 3, due to the antioxidant and anti-inflammatory properties of COS. This suggests that using CNC/COS delivery of Cur for wound healing is feasible.

#### 3.7.2. Histopathology

HE and Masson sections of day 6 and day 12 are shown in Figure 6. The wound is identified at magnifications of 40× and 100×. Granulation tissue formation, re-epithelization, collagen deposition, fibroblast cell proliferation, and angiogenesis are hallmarks for the healing of wounds [53,54,55]. HE staining indicates that the quality of granulation tissue, hair follicles, sebaceous glands, and epidermal maturation in Group I differed significantly from the other groups. On day 6, it is identified that Group I has more granulation tissue and more new capillaries and fibroblasts compared with Groups II and III. On day 12, the wounds in all groups have largely healed, while the maturity of the hair follicles and sebaceous glands in Group I are much higher than in Group II and III. Moreover, Group I also has a higher epidermal maturity.

Masson can stain collagen and muscle fibers. Collagen deposition on the wound is more intense in group I than in groups II and III on days 6 and 12. Collagen deposition and remodeling improve the tensile strength of the tissue and result in better healing [55,56]. The percentage of collagen deposition in the different groups on day 6 and 12 is presented in Figure 7A. The results of the sections indicate that the Pickering emulsion of Cur promotes capillary and fibroblast production in wounds, facilitates collagen deposition, and contributes to tissue regeneration.

#### 3.7.3. Biochemical Analysis

The ability of loaded Cur-loaded Pickering emulsion to inhibit oxidative stress during wound healing was evaluated by measuring markers of oxidative stress. SOD and MDA play important roles in the overall antioxidant defense network. MDA is formed by the reaction of free-radicals and polyunsaturated fatty acids under high oxidative stress [57,58]. The MDA content indicates the degree of oxidative damage to the tissues. SOD refers to an endogenous antioxidant that scavenges free-radicals and protects tissues by preventing lipid peroxidation [59,60]. The level of SOD can indicate the antioxidant capacity of the tissue. Figure 7B,C demonstrate the effect of Pickering emulsion loaded with Cur on different oxidative markers on days 3, 6, and 12. In the results of the current study, the level of traumatic MDA content was significantly lower in the Group I than in Groups II and III on days 3, 6, and 12. Thus, Group I and Groups II can reduce tissue damage from oxidative stress. As revealed by the findings, SOD levels were significantly higher in Group I treatment than in Groups II and III, and increased with the number of days to healing. Group II also has some antioxidant activity due to the presence of COS. Based on the results of the oxidative stress marker assay, it is known that the Cur-loaded Pickering emulsions has significantly antioxidant activity in vivo and contributes to the healing of wounds.

### 3.8. In Vitro Antibacterial

The results of the antimicrobial assay show that the saline control did not produce an antibacterial circle in either *E. coli* or *S. aureus*. As depicted in Figure 8 and Table 2 (n = 3, *p* < 0.05), both CNC/COS–Cur and CNC/COS–Oil produced zones of inhibition in *E. coli* and *S. aureus*. However, the circles produced are generally small. In Figure 8, CNC/COS–Cur produced a larger zone of inhibition than the CNC/COS group due to the loading of Cur in CNC/COS–Cur. COS is inherently antibacterial, and the antibacterial effect is correlated with its positive charge, which binds to the negative charge on the surface of bacteria and thus damages their cell membranes. The electrostatic binding of CNC to COS reduces the positive charge on the COS surface, which may reduce the inhibitory effect of COS, resulting in a small diameter of the zone of inhibition. Considering that the CNC/COS–Cur prepared has a significant slow-release effect and the release of Cur requires suitable medium, thus, the circles produced were generally small. Therefore, CNC/COS–Cur also did not produce a significant antibacterial circle.

Conventional emulsions use surfactants that can cause irritation to the skin. Pickering emulsion has advantages, such as low toxicity and adjustable permeability, without the addition of surfactants. Therefore, Pickering emulsion is more suitable for local treatment than conventional emulsion. And the release of CNC/COS–Cur was slower than that of other Cur-containing topical formulations in vitro [61]. The sustained release of Cur at the wound may facilitate better healing activities. The antioxidant properties are beneficial for wound healing. In vitro, CNC/COS–Cur showed better free-radical scavenging than the emulsions stabilized by cellulose-based nanoparticles, the Cur-containing microspheres, and nanofibrous membranes, due to the presence of COS [62,63,64].

### 3.9. Conclusions

A CNC/COS complex was prepared through electrostatic adsorption. The prepared CNC/COS formed a dense shell at the oil–water interface and could be used to stabilize 2–3 μm Pickering emulsions. CNC/COS–Cur performed well during antioxidant stress and promoted wound healing. However, as indicated by the results of in vitro antibacterial assays, it exhibited unsatisfactory antibacterial abilities, possibly because of a loss of positive COS electrical properties. Its antibacterial effect in vivo should be explored in depth. In brief, CNC/COS-prepared Pickering emulsions can be employed in the pharmaceutical industry for drug delivery.

## Figures and Tables

**Figure 1 pharmaceutics-16-01411-f001:**
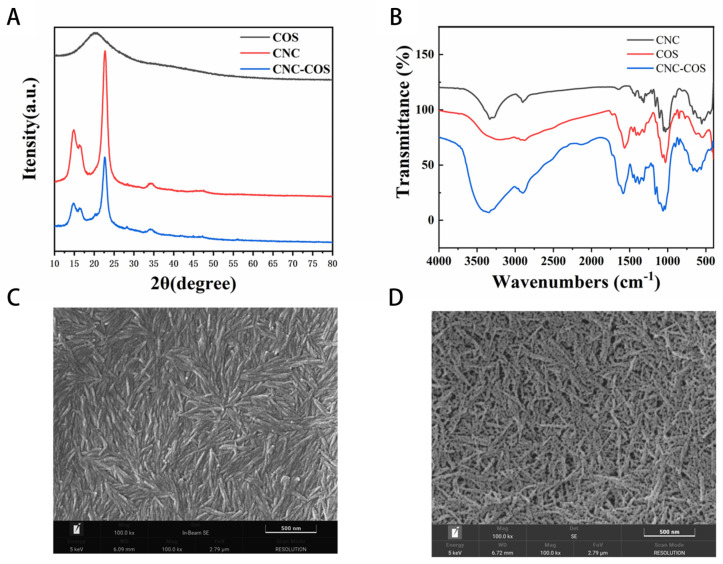
(**A**) The XRD of CNC, COS, CNC/COS. (**B**) The FTIR of CNC, COS, CNC/COS. (**C**) SEM of CN. (**D**). SEM of CNC/COS.

**Figure 2 pharmaceutics-16-01411-f002:**
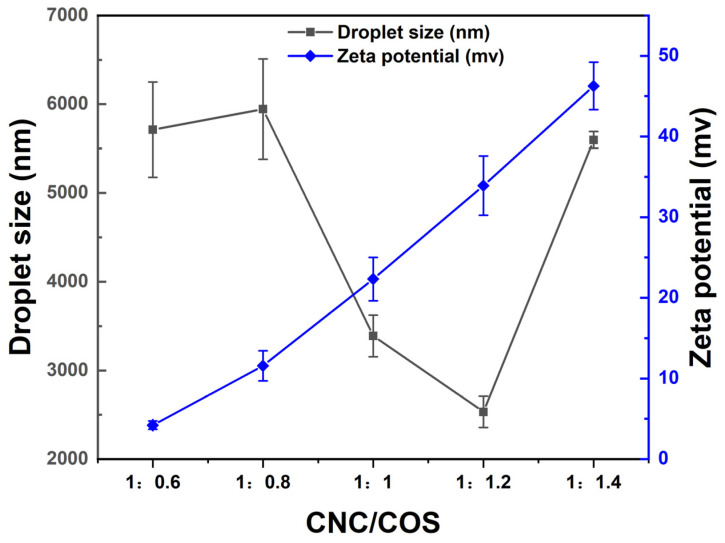
The droplet size and zeta potential of CNC/COS–Cur (Data were presented as mean ± SD, n = 3).

**Figure 3 pharmaceutics-16-01411-f003:**
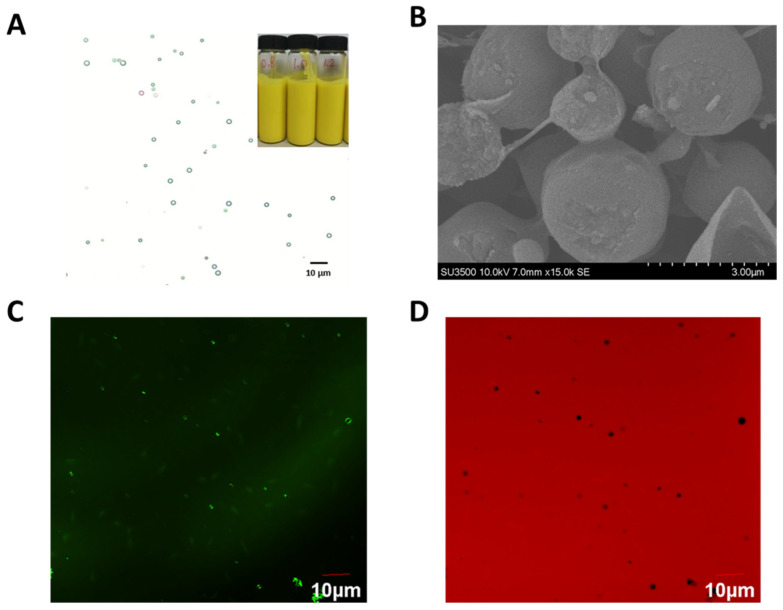
(**A**) The optical microscopy and appearance inspection of CNC/COS–Cur. (**B**) Cryo-SEM of CNC/COS–Cur. (**C**,**D**) CLSM of CNC/COS–Cur.

**Figure 4 pharmaceutics-16-01411-f004:**
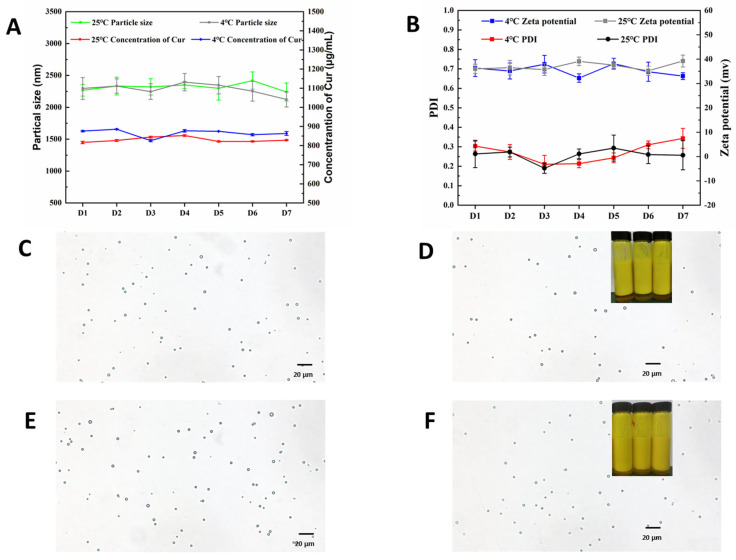
(**A**) The particle size and concentration of Cur in CNC/COS–Cur in one week. (**B**) The PDI and zeta potential of CNC/COS–Cur in one week. (**C**) The optical microscopy of CNC/COS–Cur (4 °C) in day 1. (**D**) The optical microscopy of CNC/COS–Cur (4 °C) in day 7. (**E**) The optical microscopy of CNC/COS–Cur (25 °C) in day 1. (**F**) The optical microscopy of CNC/COS–Cur (25 °C) in day 7 (Data were presented as mean ± SD, n = 3).

**Figure 5 pharmaceutics-16-01411-f005:**
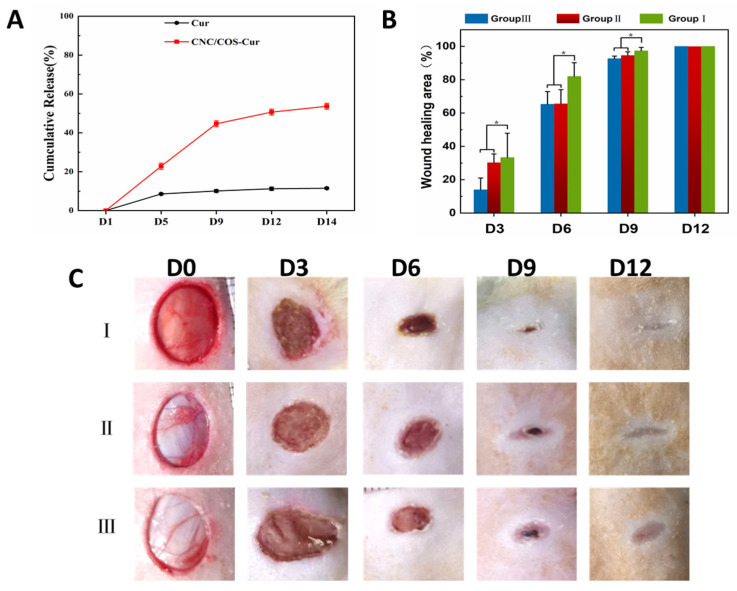
(**A**) The release profile of Cur and CNC/COS–Cur. (**B**) Wound-healing rates of CNC/COS–Cur, CNC/COS–Oil, and normal saline. (**C**) Macroscopic observations of CNC/COS–Cur, CNC/COS–Oil, and normal saline (Data were presented as mean ± SD, n = 3. * *p* < 0.05 was considered significant).

**Figure 6 pharmaceutics-16-01411-f006:**
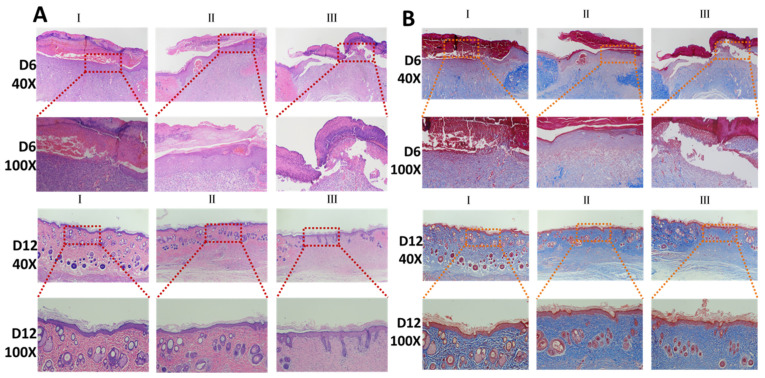
(**A**) HE sections of wounds (Group I, II, and III) on day 6 and day 12. (**B**) Masson sections of wounds (Group I, II, and III) on day 6 and day 12.

**Figure 7 pharmaceutics-16-01411-f007:**
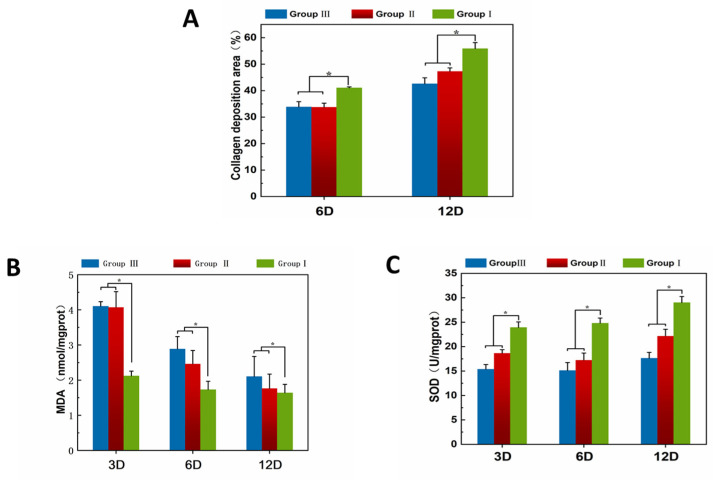
(**A**) The percentage of collagen deposition. (**B**) The MDA content in wounds. (**C**) The SOD content in wounds (Data were presented as mean ± SD, n = 3. * *p* < 0.05 was considered significant).

**Figure 8 pharmaceutics-16-01411-f008:**
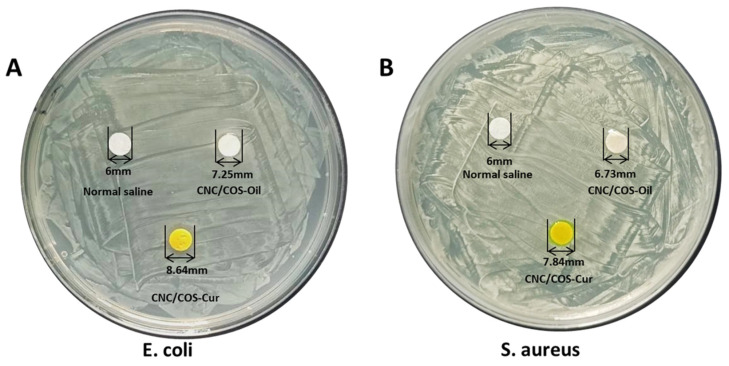
The antibacterial circle of saline, CNC/COS–Oil, CNC/COS–Cur.

**Table 1 pharmaceutics-16-01411-t001:** The statistical data for determination of Cur by proposed HPLC method.

Linearity Range, μg/mL	Coefficient of Determination (r^2^)	Regression Equations	Repeatability, RSD%
0.502–200.8	0.9998	Y = 0.0352X + 0.0395	0.14
Percentage recoveries/%, RSD/%	Precision, RSD, %	LOD, μg/mL	LOQ, μg/mL
100.3%, 0.12%	0.34%	0.40	1.02

**Table 2 pharmaceutics-16-01411-t002:** The zone of inhibition of saline, CNC/COS–Oil, CNC/COS–Cur.

	Control	CNC/COS–Oil	CNC/COS–Cur
*E. coli*	—	7.23 ± 0.47 mm	8.56 ± 0.33 mm
*S. aureus*	—	6.94 ± 0.29 mm	7.59 ± 0.22 mm

## Data Availability

Data are contained within the article.

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
