# Peer review of "Pickering Emulsion of Curcumin Stabilized by Cellulose Nanocrystals/Chitosan Oligosaccharide: Effect in Promoting Wound Healing"

_pharmaceutics, 2024, doi:10.3390/pharmaceutics16111411_

Round 1

Reviewer 1 Report

Comments and Suggestions for Authors

This is a very good and timely original article on the application of callulose and chitosan for stabilization of curcumin by means of pickering emulsion method. The manuscript fits the journal scope, the quality is adequate, although there are minor changes to be addressed prior proceeding to publication. 

As mentioned on the appropriate section, it is also suggested to improve and update the references, mainly for what concerns about cellulose, it is required to provide for a thorough insight about the polymer, its functionalization features and why this specific material was chosen (e.g. DOI: 10.1016/j.ijbiomac.2017.04.079).

Comments on the Quality of English Language

An English editing is strongly required: sentences are extra-long and a not-native English speaker could get lost. Moreover, several grammar mistakes can be retrieved, starting from the Abstract section.

Author Response

Thank you very much for taking the time to review this manuscript.

We have carefully read you recommend this article (such as DOI: 10.1016 / j.i jbiomac. 2017.04.079). We think this article fully explains the basis for our choice of cellulose and provides a good idea for the next step in our project. 

We have revised the language according to your suggestions.

We would like to thank you again for your suggestions.

Reviewer 2 Report

Comments and Suggestions for Authors

The manuscript "Pickering Emulsion of Curcumin Stabilized by Cellulose Nanocrystals/Chitosan Oligosaccharide: Effect of Promoting Wound Healing" is devoted to studying the possibility of using cellulose nanocrystals (CNC) and chitosan oligosaccharide (COS) to stabilize Pickering emulsions. This article can be accepted in the Pharmaceutics after major revision. Below are some notes.

 1. Write in the abstract the substance used as the oil phase, as well as the type of emulsion (oil-in-water).

 2. (l. 71-75) In the Introduction, much attention is paid to the characteristics and advantages of CNC and COS. However, no attention is paid to the available scientific results in the field of studying Pickering emulsions with differently charged aggregates, including for the delivery of biologically active substances. For example, in the work https://doi.org/10.21203/rs.3.rs-3873439/v1, a CNC-Chitosan stabilizer similar in composition to yours was used to stabilize the emulsion. It has been shown that a very small amount of chitosan (~1 wt%) is required to quench the negative charge of CNCs and significantly improve the stability of the emulsions.

In Introduction, pay more attention to Pickering emulsions; and in the Results section, compare your results with the above and other similar works, indicating your advantages.

 3. (l.102) Please clarify, are the weight proportions given? Why were these particular ratios chosen and the effect of small COS additions not taken into account?

 4. (l. 103) For what purpose was the freeze-drying carried out? Have you tried using samples without a drying step (aqueous dispersions)?

 5. (Section 2.2.2) The volume of the aqueous phase or the volumetric ratio of oil/water is not indicated. If I understand correctly, 20 ml of water dispersion and 10 ml of oil were used. What are the reasons for using such a ratio? Describe in your work.

 6. (l. 120-133) Why was only one emulsion sample selected for optical, confocal and electron microscopy? How suitable is the dynamic light scattering method for studying emulsions? The actual size is usually exaggerated due to droplet association.

 7. (l. 200-202) Do the components of the samples under study contribute to the determined optical density?

 8. (Section 3) It would be more logical to first present the results of the study of the stabilizer (3.2), and then of the emulsions stabilized by this stabilizer (3.1).

 9. (Section 3.1.1) At a reduced zeta potential, droplets tend to stick together, so the average hydrodynamic diameter measured by the DLS method has high values. However, this is not proof that the actual droplet size is that large. In this regard, I recommend studying a series of emulsions using optical microscopy.

 10. (Section 3.2.1) Describe in more detail the properties of CNC (crystallinity index, type I/II cellulose).

 11. (Fig. 4a) This figure shows that the average hydrodynamic diameter is approximately 2.3 μm, and in Fig. 1A – 2.5 microns. Make data consistent with each other.

 12. (Fig. 4C-F). Show in the figures histograms of droplet distributions with mean and standard deviation. Describe the results in the text of the manuscript and compare them with DLS.

 13. (Section 3.6) Give a mathematical interpretation of the release kinetics. To more adequately display the release curve (Fig. 5A), construct a graph in normal coordinates (the x axis will have equal segments between days (1, 2, 3...).

 14. (l. 392-395) How do you propose to thicken the shell of the drop and thereby control the release?

Some other notes:

 l. 71 “adding an oppositely charged electrolyte to the aqueous phase.” Not quite the right phrase. The electrolyte dissociates into cations and anions and cannot be charged.

 l.90 " 5x10 KDa". Fix it.

 l. 118, 186 Most likely you need to replace “dynamic light diffraction” with “dynamic light scattering”.

 l. 128 decipher «FIT».

 l. 142, 143 “Diffraction X-ray Diffraction (XRD)”, “Diffraction XRD”. Fix it.

 l. 148 Thermo fisher,America -> Thermo Fisher, USA

 l. 155 Tesken -> Tescan

 l. 158 "morphology was identified at 70k 150k". Fix it.

 l. 185, 187 "2.2.3.1 and 2.25". These sections are not in your work.

 l. 188 "on\e week". Fix it.

l. 290 Describe the CNC:COS ratio in the figure caption.

l. 388 Please clarify in the section that the MCT solution of Cur is used.

Author Response

Thank you very much for taking the time to review this manuscript.

We are very grateful for your recognition of our work and your suggestions for this article.

Regarding your suggestion, we have made changes in the article and here is our response for your review.

  1. Write in the abstract the substance used as the oil phase, as well as the type of emulsion (oil-in-water).

Author's Response: We have written the substances used as oil phase and the type of emulsion (oil-in-water) in the summary as you requested.

  1. (l. 71-75) In the Introduction, much attention is paid to the characteristics and advantages of CNC and COS. However, no attention is paid to the available scientific results in the field of studying Pickering emulsions with differently charged aggregates, including for the delivery of biologically active substances. For example, in the work https://doi.org/10.21203/rs.3.rs-3873439/v1, a CNC-Chitosan stabilizer similar in composition to yours was used to stabilize the emulsion. It has been shown that a very small amount of chitosan (~1 wt%) is required to quench the negative charge of CNCs and significantly improve the stability of the emulsions.

In Introduction, pay more attention to Pickering emulsions; and in the Results section, compare your results with the above and other similar works, indicating your advantages.

Author's Response: We have been in the process of revamping our Introduction section and have given more information about the therapeutic effects and applications of Pickering emulsions and indicated our advantages.

  1. (l.102) Please clarify, are the weight proportions given? Why were these particular ratios chosen and the effect of small COS additions not taken into account?

Author's Response: We pre-experimented with some COS concentrations and finalized a COS concentration of 1.2 wt% (compared to CNC). The ratio of CNC and COS is the most suitable ratio screened in the preliminary formulation experiments, taking into account the stability of the Pickering emulsion, the size of the particle size, the drug loading capacity and other factors.

  1. (l. 103) For what purpose was the freeze-drying carried out? Have you tried using samples without a drying step (aqueous dispersions)?

Author's Response: We can directly observe the surface contour, morphology of the milk droplets through the scanning electron microscope, which is really missing in the experimental design. The reason for choosing freeze-drying is that after freeze-drying, the droplets can not only maintain their original physical form and functional state, but also facilitate the clear observation of their macromolecular three-dimensional structure. Therefore, the main purpose of using cryo SEM is to allow a clearer observation of the structure of the shell layer formed by CNC/COS in Pickering's emulsion.

  1. (Section 2.2.2) The volume of the aqueous phase or the volumetric ratio of oil/water is not indicated. If I understand correctly, 20 ml of water dispersion and 10 ml of oil were used. What are the reasons for using such a ratio? Describe in your work.

Author's Response: Yes, 10 ml of drug-loaded oil phase was dispersed in 20 ml of aqueous phase. As in the third point, at the very early stage of the experiment, we screened this Pickering emulsion dosage form for formulation preparation methods, taking into account the stability of the emulsion, the size of the particle size, the amount of drug loaded, and other influencing factors to screen for the most suitable oil to water ratio (1:2).

  1. (l. 120-133) Why was only one emulsion sample selected for optical, confocal and electron microscopy? How suitable is the dynamic light scattering method for studying emulsions? The actual size is usually exaggerated due to droplet association.

Author's Response: The DLS method can visualize the average value of the obtained particle size and the distribution of the particle size, we diluted the droplets in the experiment and tried to disperse the droplets as much as possible and then measured them, but we did not take the aggregation of the droplets into account, thank you for your comments, we will consider the DLS method in combination with the observation of the droplets by microscope in the future.

  1. (l. 200-202) Do the components of the samples under study contribute to the determined optical density?

Author's Response: The composition of the studied samples had no influence on the determined optical density.

  1. (Section 3) It would be more logical to first present the results of the study of the stabilizer (3.2), and then of the emulsions stabilized by this stabilizer (3.1).

Author's Response: Thank you for your suggestion, we have made the changes in the article as you requested.

  1. (Section 3.1.1) At a reduced zeta potential, droplets tend to stick together, so the average hydrodynamic diameter measured by the DLS method has high values. However, this is not proof that the actual droplet size is that large. In this regard, I recommend studying a series of emulsions using optical microscopy.

Author's Response: We thank the reviewers for their suggestion and we will add the optical microscopy method for size measurements in subsequent studies.

  1. (Section 3.2.1) Describe in more detail the properties of CNC (crystallinity index, type I/II cellulose).

Author's Response: Our conclusions on crystallinity and type I cellulose were calculated from XRD diffraction peak raw data and have been changed in the article.

  1. (Fig. 4a) This figure shows that the average hydrodynamic diameter is approximately 2.3 μm, and in Fig. 1A – 2.5 microns. Make data consistent with each other.

Author's Response: Thanks to the opinions of the referees, we figure 1 a is actually 2.3 mu m, figure 4 a is 2.3 microns, appeared in the paper we edit the mistake. We have made changes in the article.

  1. (Fig. 4C-F). Show in the figures histograms of droplet distributions with mean and standard deviation. Describe the results in the text of the manuscript and compare them with DLS.

Author's Response: The line graph in Figure 1 has the grain size distribution for the corresponding period.

  1. (Section 3.6) Give a mathematical interpretation of the release kinetics. To more adequately display the release curve (Fig. 5A), construct a graph in normal coordinates (the x axis will have equal segments between days (1, 2, 3...).

Author's Response: The results of the cumulative release of CNC/COS-Cur and Cur are shown in the figure. 10% of the oil solution of Cur was released cumulatively over 14 days, while the total release of CNC/COS-Cur was about 55%. The release of Cur from CNC/COS-Cur was mainly concentrated in the first 9 days and the release appeared to reach a plateau at day 10. The cumulative release from CNC/COS-Cur was more than five times higher than that from the Cur oil solution alone.

  1. (l. 392-395) How do you propose to thicken the shell of the drop and thereby control the release?

Author's Response: The diffusion of CUR from the oil phase to the water phase needs to pass through the shell of CNC/COS, which acts as a stabilizer to separate the water from the two phases, so we believe that the thicker the shell layer, the longer the diffusion of the drug from the oil phase to the water phase takes to achieve the purpose of slow release.

Some other notes:

  1. 71 “adding an oppositely charged electrolyte to the aqueous phase.” Not quite the right phrase. The electrolyte dissociates into cations and anions and cannot be charged.

 l.90 " 5x10 KDa". Fix it.

  1. 118, 186 Most likely you need to replace “dynamic light diffraction” with “dynamic light scattering”.
  2. 128 decipher «FIT».
  3. 142, 143 “Diffraction X-ray Diffraction (XRD)”, “Diffraction XRD”. Fix it.
  4. 148 Thermo fisher,America -> Thermo Fisher, USA
  5. 155 Tesken -> Tescan
  6. 158 "morphology was identified at 70k 150k". Fix it.
  7. 185, 187 "2.2.3.1 and 2.25". These sections are not in your work.
  8. 188 "on\e week". Fix it.
  9. 290 Describe the CNC:COS ratio in the figure caption.
  10. 388 Please clarify in the section that the MCT solution of Cur is used.

Author's Response: Thank you for your careful review, we have made the changes you requested in the article.

Again, we appreciate your review and suggestions!

Reviewer 3 Report

Comments and Suggestions for Authors

The paper describes the loading of curcumin into particles prepared by pickering emulsion of a mixture of chitosan and cellulose nanocrystals. Pickering emulsions are emulsions stabilizing the interface by a particle rather than by a classical surfactant.

The preparation of pickering emulsions based on chitosan is not possible due to too high hydrophilicity. It is reported in the state of the art that the addition of a polyanion counterbalancing the cationic sites of chitosan is a trick to reach these pickering emulsions. In this new paper, the authors decided to use cellulose nanocrystals for that sake.

The formulation of particles loaded by curcumin prepared by pickering emulsion based on chitosan is already reported by using other polymers as polyanions. These papers should be clearly cited in the introduction. One of them is cited [REF 31 - Environmental stability and curcumin release properties of Pickering emulsion 580 stabilized by chitosan/gum arabic nanoparticles - Int J Biol Macromol - 157 [2020] - 202-211] but for other reasons, which is not sufficient.

Nevertheless, compared to previous studies, the authors bring important results on wound healing. The biological study is important to bring the originality necessary for publication. The introduction focusses quasi exclusively on the formulation by pickering emulsions and just one sentence at the end of the introduction for the wound healing application. I have the feeling that the biological part could be better highlighted in the introduction because the paper is not just a paper on pickering emulsion but on the use of pickering emulsions for a biomedical application.

The authors should be able to revise the introduction of their paper to take into account my remarks and I recommend to publish the paper with minor changes.

Author Response

Thank you very much for taking the time to review this manuscript.

We have added a promising description of the application of pickering emulsions in wound healing in the introduction. And based on your suggestion, we have a clear direction for the next step of our research, and we will continue to investigate the biomedical applications of pickering emulsions.

Thanks again for your suggestions.

Round 2

Reviewer 2 Report

Comments and Suggestions for Authors

  The authors have made some corrections to the manuscript. However, some points are not reflected in the text of the manuscript.

1. I did not find any indication of the oil phase used in the abstract.

2. I did not find any comparison of the physicochemical properties of the emulsions with other works.

5. Indicate the oil to water ratio (1:2) in the manuscript

13. Apparently, the authors did not quite understand the request correctly. There are various mathematical models describing the release kinetics. Analyze the data obtained.

Author Response

1.We introduced the substances in the oil phase in the preparation method (2.2.2) in the main text, and we did not emphasize the substances in the oil phase in the abstract because we thought that the oil could be replaced using other substances.

2.Dear reviewer, as you said, the emulsions prepared in this study are only preparation process optimisation and some potency exploration, there is no comparison with other works, which is something we need to correct subsequently, thank you for your suggestion.

5.Thanks for the reminder, we have added the ratio of oil phase to water phase (1:2) in (2.2.2) of the article.

13.In fact, we detected the cumulative release on days 3 and 6 in our pre-experiment, but because the levels were lower on the first 3 days, we directly measured the cumulative release on day 5 in the formal experiment. Because of time and sample problems, we may not be able to add this part of the data in time, but please understand that this will be the direction of our attention in the subsequent research.